# Maintaining resting cardiac fibroblasts *in vitro* by disrupting mechanotransduction

George Gilles[1], Andrew D. McCulloch[1,2], Cord H. Brakebusch[3], Kate M. Herum[3]*

**1** Department of Bioengineering, University of California San Diego, La Jolla, California, United States of America, **2** Department of Medicine, University of California San Diego, La Jolla, California, United States of America, **3** Biotech Research and Innovation Centre, University of Copenhagen, Copenhagen, Denmark

* kate.herum@bric.ku.dk

**Data Availability Statement:** All relevant data are within the manuscript.

## Abstract

Mechanical cues activate cardiac fibroblasts and induce differentiation into myofibroblasts, which are key steps for development of cardiac fibrosis. *In vitro*, the high stiffness of plastic culturing conditions will also induce these changes. It is therefore challenging to study resting cardiac fibroblasts and their activation *in vitro*. Here we investigate the extent to which disrupting mechanotransduction by culturing cardiac fibroblasts on soft hydrogels or in the presence of biochemical inhibitors can be used to maintain resting cardiac fibroblasts *in vitro*. Primary cardiac fibroblasts were isolated from adult mice and cultured on plastic or soft (4.5 kPa) polyacrylamide hydrogels. Myofibroblast marker gene expression and smooth muscle α-actin (SMA) fibers were quantified by real-time PCR and immunostaining, respectively. Myofibroblast differentiation was prevented on soft hydrogels for 9 days, but had occurred after 15 days on hydrogels. Transferring myofibroblasts to soft hydrogels reduced expression of myofibroblast-associated genes, albeit SMA fibers remained present. Inhibitors of transforming growth factor β receptor I (TGFβRI) and Rho-associated protein kinase (ROCK) were effective in preventing and reversing myofibroblast gene expression. SMA fibers were also reduced by blocker treatment although cell morphology did not change. Reversed cardiac fibroblasts maintained the ability to re-differentiate after the removal of blockers, suggesting that these are functionally similar to resting cardiac fibroblasts. However, actin alpha 2 smooth muscle (Acta2), lysyl oxidase (Lox) and periostin (Postn) were no longer sensitive to substrate stiffness, suggesting that transient treatment with mechanotransduction inhibitors changes the mechanosensitivity of some fibrosis-related genes. In summary, our results bring novel insight regarding the relative importance of specific mechanical signaling pathways in regulating different myofibroblast-associated genes. Furthermore, combining blocker treatment with the use of soft hydrogels has not been tested previously and revealed that only some genes remain mechano-sensitive after phenotypic reversion. This is important information for researchers using inhibitors to maintain a "resting" cardiac fibroblast phenotype *in vitro* as well as for our current understanding of mechanosensitive gene regulation.

**Funding:** K.M.H. has received funding from the European Union's Horizon 2020 research and innovation programme under the Marie Sklodowska-Curie grant agreement No 795390. https://ec.europa.eu/programmes/horizon2020/en/ The project was supported in part by NIH grants 1 R01 HL137100-01, U01 H127654 and 1 U01 HL126273. https://www.nih.gov/grants-funding The funders had no role in study design, data collection and analysis, decision to publish, or preparation of the manuscript.

**Competing interests:** A.D.M. is a co-founder of and has an equity interest in Insilicomed Inc. and an equity interest in Vektor Medical, Inc. He serves on the scientific advisory board of Insilicomed, and as scientific advisor to both companies. Some of his research grants have been identified for conflict of interest management based on the overall scope of the project and its potential benefit to these companies. The author is required to disclose this relationship in publications acknowledging the grant support; however, the research subject and findings reported in this study did not involve the companies in any way and have no relationship with the business activities or scientific interests of either company. The terms of this arrangement have been reviewed and approved by the University of California San Diego in accordance with its conflict of interest policies. We declare that the competing interest statements by A.D.M. does not alter our adherence to PLOS ONE policies on sharing data and materials.

## Introduction

Activation of cardiac fibroblasts is a key step in development of cardiac fibrosis. In response to increased mechanical stress caused by increased left ventricular pressure or myocardial stiffening, cardiac fibroblasts differentiate into myofibroblasts characterized by de novo gene expression of actin alpha 2 smooth muscle (Acta2) and assembly of contractile smooth muscle α-actin (SMA) fibers, as well as enhanced production of typical cardiac extracellular matrix (ECM) genes including collagen (Col) 1a1, 1a2, 3a1 and the collagen cross-linking enzymes lysyl oxidase (Lox) [1] and Lox-like 2 [2]. In addition, expression of connective tissue growth factor (Ctgf) [3] and periostin (Postn) [4] is associated with the myofibroblasts phenotype. *In vivo*, the increases in myofibroblast markers are accompanied by decreased expression of transcription factor 21 (Tcf21), a marker of resting cardiac fibroblasts [4]. These changes in cardiac fibroblast phenotype alter ECM composition and structure leading to increased stiffness of the heart, which can compromise diastolic function. Importantly, studying cardiac fibroblast activation *in vitro* is hampered by rapid myofibroblast differentiation in response to the high stiffness of plastic culturing conditions [1]. We here investigate to what extent disrupting mechanotransduction can be used to maintain resting cardiac fibroblasts *in vitro*, thereby enabling the study of biomechanical activation of resting cardiac fibroblasts.

Soft hydrogels can be used as culturing substrate to eliminate the effect of high substrate stiffness on cardiac fibroblast activation [1, 5]. However, the use of soft hydrogels have some limitations including small sample size and the need for frequent, time-consuming isolation of primary cells for direct plating onto hydrogels. Furthermore, it is not known for how long cardiac fibroblasts remain inactivated when cultured on hydrogels. We have previously used soft hydrogels to maintain resting cardiac fibroblasts for up to 5 days [5], but some experiments, such as CRISPR/Cas9 gene editing requires longer cell culture durations. Here we examine the time frame for maintaining resting cardiac fibroblasts on soft hydrogels.

Instead of changing the mechanical environment, it may be possible to transiently inhibit the cardiac fibroblast's ability to sense mechanical stress and thereby render cardiac fibroblasts unresponsive to the stiff *in vitro* culturing conditions. Although ECM gene expression was not examined, culturing cardiac fibroblasts in the presence of a transforming growth factor β receptor (TGFβR) inhibitor was previously shown to prevent stiffness-induced development of SMA fibers [6]. However, experiments required continuous culturing with TGFβR inhibitor which would interfere with subsequent experiments studying cardiac fibroblast activation.

Several other mechanotransduction signaling pathways are known to regulate cardiac fibroblast activation [7, 8]. We and others have previously shown that calcineurin-nuclear factor of activated T cells (NFAT) signaling is important for myofibroblast differentiation [9, 10], and that Rho-associated protein kinase (ROCK) regulates the myofibroblast-associated transcription factor myocardin-related transcription factor A (MRTF-A) by altering cytoskeletal dynamics [11]. Although it is known that these pathways are important for myofibroblast differentiation, their relative contributions and the effect of dual inhibition have not previously been tested. Here we examine whether single or combined treatment with TGFβRI, ROCK and calcineurin inhibitors can prevent and/or reverse myofibroblast marker expression. Furthermore, we test whether "reversed" cells have preserved mechanosensitivity, and whether culturing "reversed" cardiac fibroblast on hydrogels may serve as a method for studying resting cardiac fibroblasts *in vitro*, even after prolonged culturing time and multiple passages on plastic.

## Methods

### Cardiac fibroblast isolation and culturing

The project was approved by the veterinarian at Department of Experimental Medicine, University of Copenhagen (project license #P18-289). Mice were briefly anesthetized with isoflurane (open drop) and euthanized by cervical dislocation. Cardiac fibroblasts were isolated from adult mice (C56Bl6, Taconic, Denmark) as previously described [5]. Cells were expanded on plastic and treated with Y27632 (25μM), SB431542 (10μM) and cyclosporine A (CsA;1μM) for 3 days, adding fresh blocker solutions every 24h, or plated directly onto soft collagen I-coated polyacrylamide hydrogels with a stiffness of 4.5 kPa that were generated as previously described [12].

### Immunostaining

Cardiac fibroblasts were fixed in 4% PFA, permeabilized with 0.1% triton, quenched with 25 mM glycine and blocked in 5% goat serum. Primary and secondary antibodies were diluted in 2% goat serum. Mouse anti-SMA antibody (1A4 clone, # 14-9760-82, Thermo Fisher, Denmark) was diluted 1:1000 and 488-Alexa secondary antibody (anti-mouse) was diluted 1:2000. SMA was quantified by taking the average GFP intensity for cells on 10 micrographs per cover slip, three cover slips (N = 3) per condition. The same exposure time and gain was used for all images and quantification was performed using Cell Profiler [13]. The "No 1AB negative control" was used to determine suitable exposure time (no signal in No 1AB control). Values were normalized to non-treated controls for each round of experiments. The experiments were done with cardiac fibroblasts from two separate isolations.

### RNA isolation, cDNA synthesis and real-time PCR

RNA isolation was performed using GenElute Mammalian Total RNA Miniprep Kit (Sigma), and cDNA using the TaqMan Reverse Transcription Reagents kit (#N8080234, Applied Biosystems). Nanodrop was used to determine concentration and quality of RNA. A "no template negative control" and "no reverse transcriptase (RT) negative control" was included in the cDNA synthesis and subsequent RT-PCR runs. Real-time PCR was performed using SYBR Green and primers for collagen (Col) 1a1, Col3a1, Acta2, connective tissue growth factor (Ctgf), lysyl oxidase (Lox), periostin (Postn), transcription factor 21 (Tcf21). mRNA was normalized to housekeeping gene ribosomal 18S. Since one Ct value reflects a doubling of mRNA, Ct values were transformed to a linear scale by exponentiation. Ct levels are inversely proportional to the amount of target nucleic acid in the sample; therefore, ΔCt is a negative value. The data presented in Fig 2F were presented as $2^{-\Delta Ct}$. All other mRNA data were presented as relative to the control group.

### Statistics

Normal distribution was determined by D'Agostino & Pearson test. For comparisons of multiple groups, one-way ANOVA with Dunnett's *post-hoc* test was used. For two-group comparisons Student's *t*-test was used (two-tailed) andcorrected for multiple comparisons using Holm-Sidak test. *p<0.05, **p<0.01, ***p<0.001, not significant (n.s.).

## Results

### The myofibroblast phenotype peaks at day nine of culture on plastic

To determine the effect of a stiff culturing substrate on myofibroblast differentiation, we measured the expression of myofibroblast-associated genes at different time points after plating

cells on plastic. As expected, we observed a rapid increase in the classical myofibroblast marker *Acta2* mRNA (Fig 1A) and protein (Fig 1B) in response to plastic culture conditions, resulting in the formation of clear SMA fibers in the cytoplasm (Fig 1C). Acta2 mRNA and SMA staining intensity peaked at 9 days where after it declined. Expression of Col1a1 and Lox also increased during culture on plastic (Fig 1D and 1E), but started to decline after 12 days, indicating that myofibroblasts do not comprise a permanently differentiated state.

## Culturing cardiac fibroblasts on soft gels delays myofibroblast differentiation

We previously showed that culturing cardiac fibroblasts on soft hydrogels, mimicking the stiffness of the healthy heart, prevented myofibroblast differentiation over the course of several days (5). To examine the time frame of this effect, we examined marker gene expression after plating cardiac fibroblasts directly onto soft (4.5kPa) hydrogels and culturing them for 3, 9 and 15 days (Fig 2A).

SMA staining intensity was weak in cardiac fibroblasts cultured on soft gels for 3 and 9 days, but strong after 15 days on gels (Fig 2B). Cardiac fibroblasts cultured for 12 days on soft gels also showed development of SMA fibers in some cells, but not all (data not shown), suggesting that the specific time point for differentiation may vary among cells. As expected, low stiffness conditions were accompanied by reduced expression of myofibroblast markers Acta2, Col1a1 and Ctgf at 3 and 9 days, and Lox at 3 days when compared to cells cultured on plastic (Fig 2C). In contrast, Col3a1 mRNA was unaltered at these time points, suggesting that regulation of Col3a1 is not regulated by stiffness. Importantly, myofibroblast marker expression was

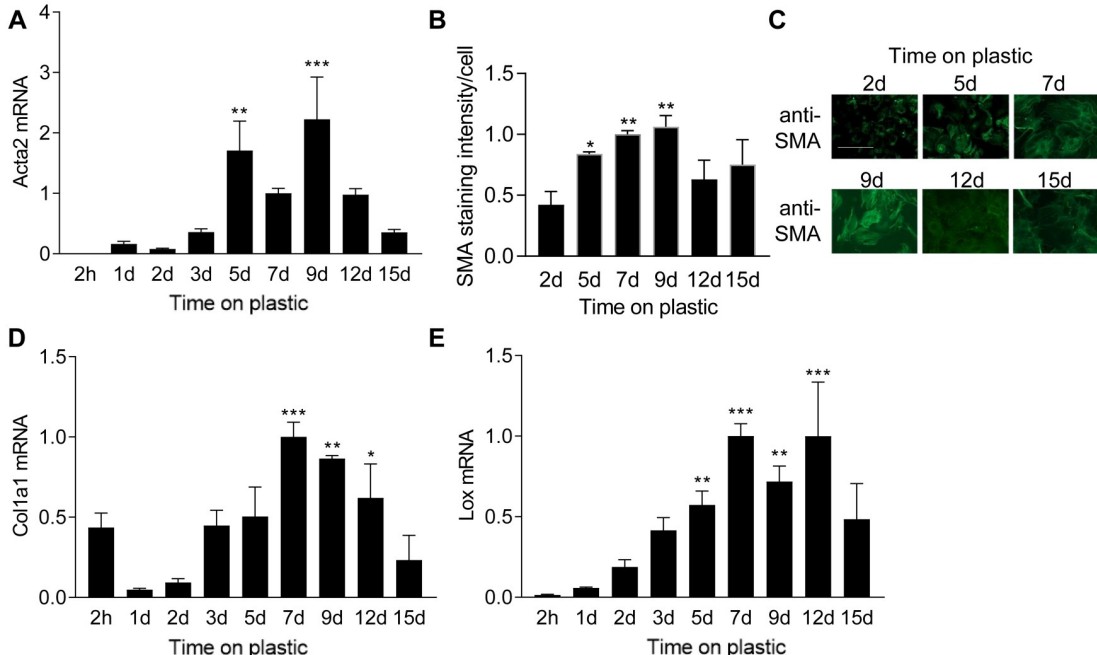

**Fig 1. Cardiac fibroblast differentiation during culture on plastic.** Actin alpha 2 smooth muscle (Acta2) mRNA (A) and smooth muscle α-actin (SMA) staining intensity (B) in cardiac fibroblasts plated on plastic for different times. C) Representative immunostaining images for SMA. Scale bar 100μm. D) mRNA expression of collagen (Col) 1a1 and lysyl oxidase (Lox) in cardiac fibroblast cultured on plastic for different durations of time. One-way ANOVA with Dunnett's multiple comparisons test as indicated for significant differences compared to the 1 day time point. N = 3 (2h, 1d, 2d, 3d, 9d, 12d and 15d), N = 5 (5d), N = 9 (7d). mRNA was normalized to ribosomal 18S.

no longer reduced in cardiac fibroblasts on soft gels compared to plastic after 15 days, and Col3a1 was even 8-fold increased (Fig 2C). These data show that myofibroblast differentiation can be delayed, but not prevented, by using soft hydrogels as the substrate.

Transferring myofibroblasts from plastic to soft gels (Fig 2D) did not reduce SMA fibers (Fig 2E), and surprisingly, Acta2 mRNA was significantly increased (Fig 2F), suggesting that there may be a temporal window during which Acta2 expression can be reversed. In contrast to Acta2, the remaining myofibroblast markers were markedly decreased after four days on soft gels, suggesting that once the myofibroblast phenotype is established, a stiff substrate is necessary for continued expression of many myofibroblast-associated genes.

## Inhibiting ROCK and TGFβRI is most efficient for preventing and reversing the myofibroblast phenotype

We then tested the effect of blocking mechanotransduction signaling pathways (Fig 3A) by using single and combined treatments with the ROCK inhibitor Y27632 (Y27), the TGFβRI

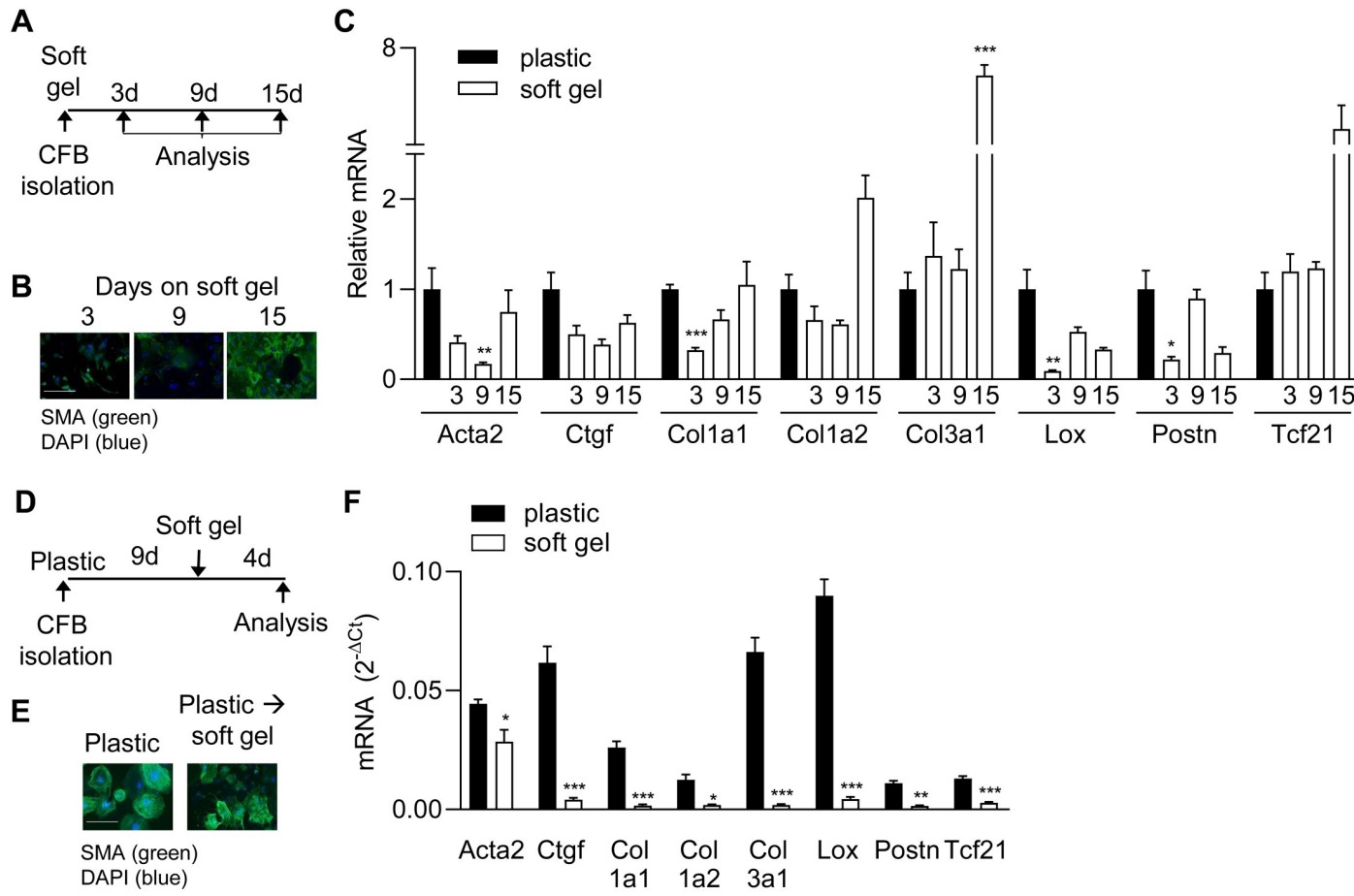

**Fig 2. Relieving cells of high stiffness delays, but does not prevent, myofibroblast differentiation.** A) Schematic illustration of protocol used for experiments in B and C. Cardiac fibroblasts were cultured on plastic or soft gels for 3, 9 or 15 days (d) and then analysed. B) Immunostaining for smooth muscle α-actin (SMA; green) in cardiac fibroblasts. Nuclei were stained with DAPI (blue). Scale bar 100μm. C) mRNA expression of myofibroblast markers and Tcf21 at 13 days after seeding. Y-axis value, $2^{-\Delta Ct}$, represents mRNA levels for each gene relative to 18S rRNA. ΔCt is the Ct value of the gene of interest minus the Ct value of 18S rRNA. D) Schematic illustration of protocol used for experiments in E and F. Cardiac fibroblasts were cultured on plastic for 9d where after they were either transferred to soft gels of kept on plastic for an additional 4d. E) Immunostaining for SMA (green) in cardiac fibroblasts. Nuclei were stained with DAPI (blue). Scale bar 100μm. F) mRNA expression of myofibroblast markers and Tcf21. Significant differences were determined by Student's *t*-test corrected for multiple comparisons using Holm-Sidak test. N = 8 (C, plastic and 3d N = 4 (C, 9d), N = 3 (C, 15d) and N = 3 (F).

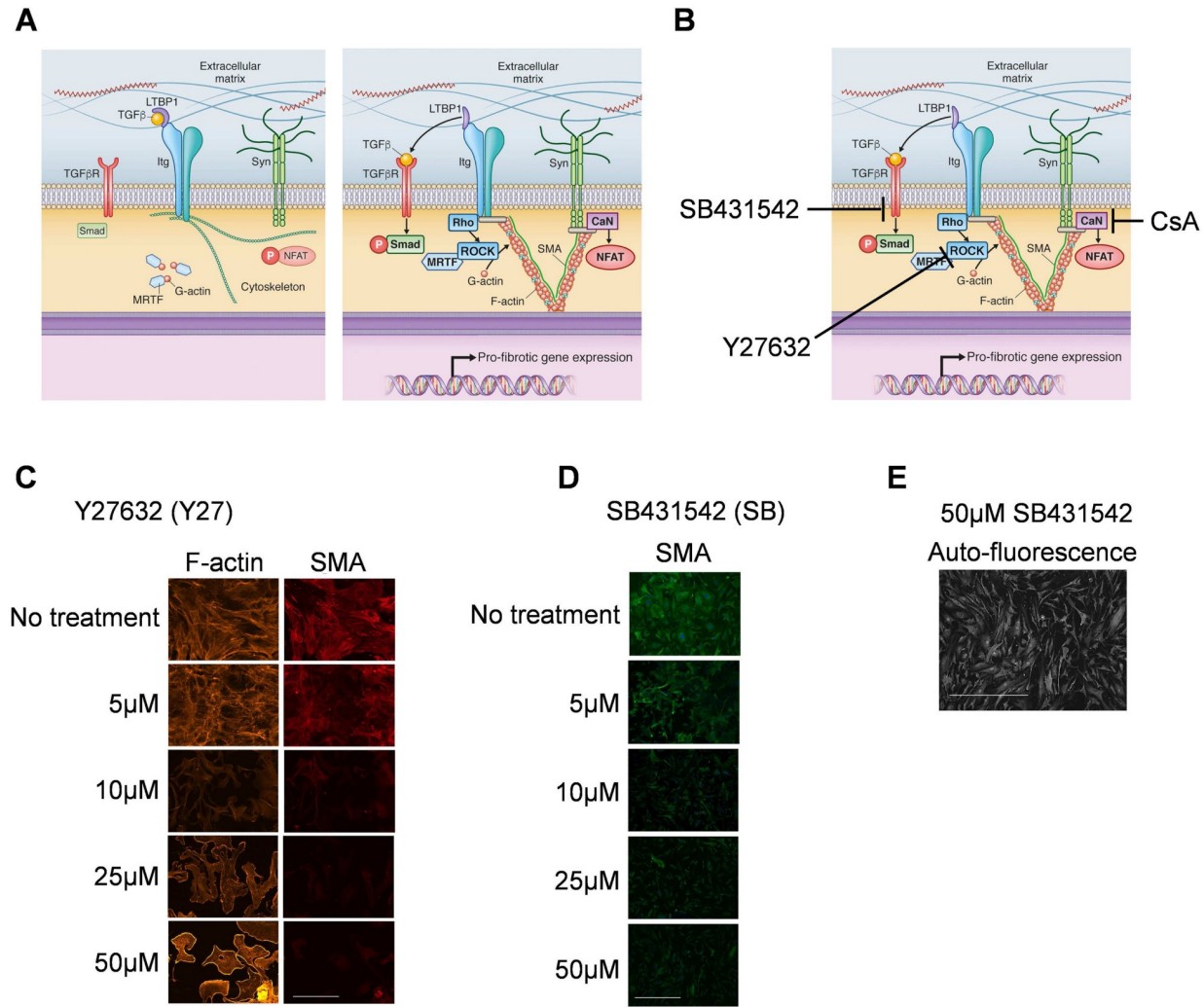

**Fig 3. Effect of different concentration of ROCK and TGFβRI inhibitors on smooth muscle α-actin fibers.** A) Schematic diagram (modified from Herum et al., *JCM*, 2017) of the three main mechanically activated signalling pathways that lead to myofibroblast differentiation and pro-fibrotic gene expression. B) Inhibition points using the inhibitors Y27632, SB431542 and cyclosporine A (CsA). Immunostaining of cardiac fibroblasts for smooth muscle α-actin (SMA) in cardiac fibroblasts treated with different concentrations of Y27632 (C) and SB431542 (D and E). The micrograph in E is the same as in D. Phalloidin was used to stain F-actin (C) and DAPI to stain nuclei (D and E). Scale bar 100μm (C) and 200μm (D and E).

inhibitor SB431542 (SB), and the calcineurin inhibitor CsA (Fig 3B). To determine optimal blocker concentrations, we studied responses in SMA fiber formation. Concentrations >10 μmol/L for both Y27 and SB efficiently reduced SMA fiber formation, suggesting inhibition of mechanotransduction (Fig 3C and 3D). Since the half-life for Y27 has been reported to be 12–16 h [14], we used a concentration of 25 μmol/l for Y27. Interestingly, SB concentrations higher than 10μmol/l induced a different morphology compared to lower concentrations (Fig 3E) which did not resemble typical fibroblast morphology. Therefore, we used a concentration of 10 μmol/l for SB, and replaced blocker solutions every 24 h. CsA was applied at a final concentration of 1 μmol/L according to our previous publication [9].

The protocols used to test if blockers could ***prevent*** or ***reverse*** myofibroblast differentiation are illustrated in Fig 4A and 4B. Inhibiting ROCK or TGFβRI with Y27 and SB, respectively,

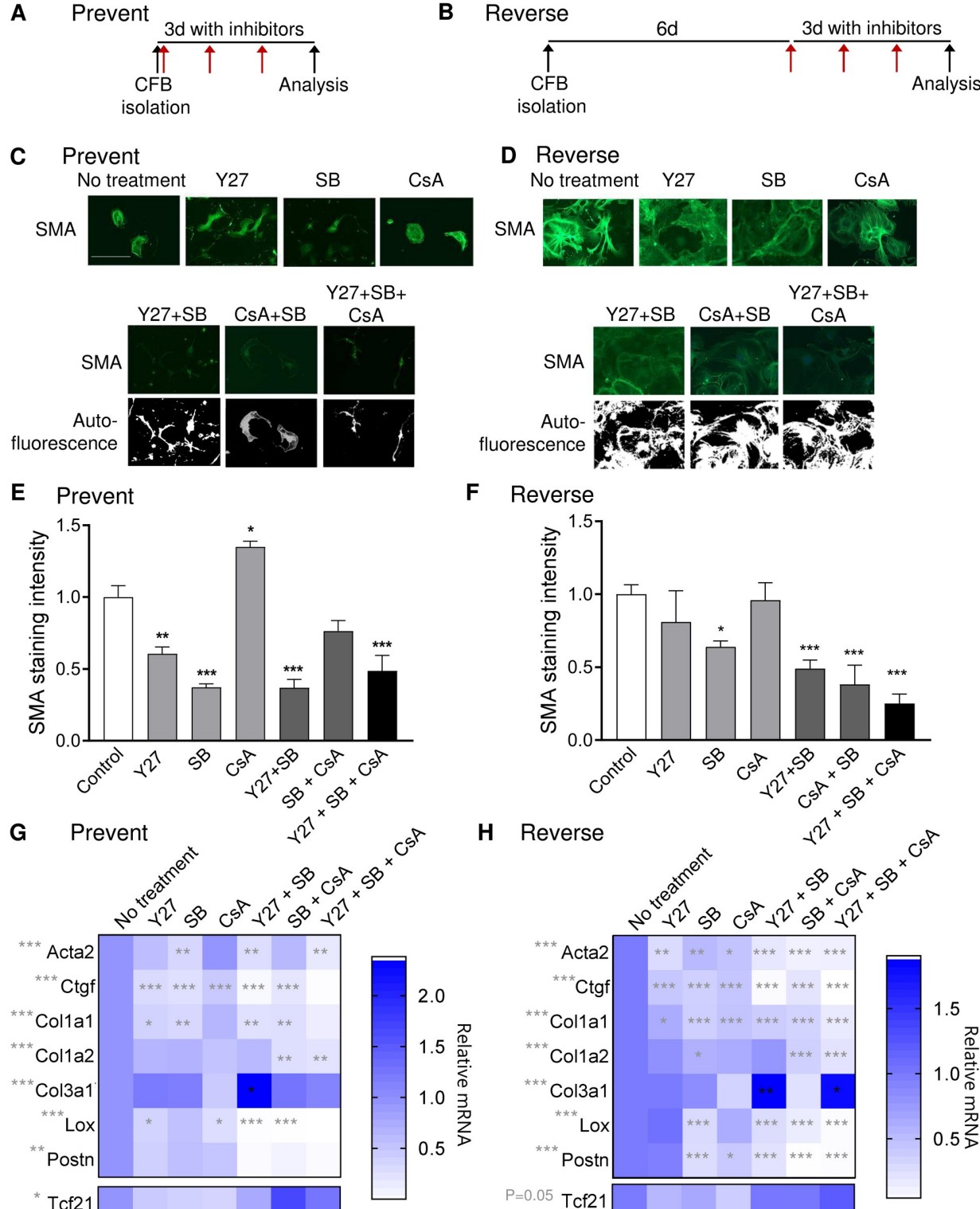

**Fig 4. Inhibiting ROCK and TGFβRI is most efficient for preventing and reversing the myofibroblast phenotype.** A and B) Schematic illustration of treatment regimen. Red arrows indicate the addition of fresh inhibitor solution every 24h. C and D) Immunostaining for smooth muscle α-actin (SMA) in cardiac fibroblasts treated with blockers of ROCK (Y27), TGFβRI (SB) and calcineurin (CsA). Scale bar 100μm. E and F) Quantification of SMA staining intensity. G and H) Expression of myofibroblast markers actin alpha 2 smooth muscle (Acta2), connective tissue growth factor (Ctgf), collagen (Col) 1a1, Col1a2, Col3a1, lysyl oxidase (Lox) and periostin (Postn), and the marker of resting cardiac fibroblasts, transcription factor 21 (Tcf21). One-way ANOVA as indicated beside gene name and Dunnett's multiple comparisons test for significant differences compared to the "no treatment" control as indicated in the heat map. N = 3 (E and F), 4 (G) 4–7 (H).

*prevented* SMA expression and fiber formation, and altered cell morphology into smaller cells (Fig 4C and 4E). Y27 also induced a more dendritic appearance compared to the other groups, resembling the morphology of cardiac fibroblasts cultured on soft gels. Surprisingly, calcineurin inhibition with CsA increased staining intensity and had no effect on cell morphology. Combining Y27 and SB reduced SMA to the same extent as SB alone, however, cell morphology resembled cells treated with Y27 inhibitor alone. Combined treatment with all three inhibitors did not further reduce SMA compared with Y27 and SB co-treatment, and cells appeared smaller and more compact, possibly indicating reduced cell viability or proliferation in this group (Fig 4C).

For the *reverse* experiment, SB reduced SMA staining intensity, and this effect was augmented by adding Y27 or CsA (Fig 4D and 4F). No clear differences in cell morphology were observed for the reversed cardiac fibroblasts (Fig 4D), indicating that inhibiting mechanotransduction signaling at this time point is not sufficient to reverse cell morphological phenotype.

SB *prevented* Acta2 upregulation, confirming that TGFβRI signaling is essential for myofibroblast differentiation (Fig 4G). However, including Y27 additionally prevented induction of Lox and Postn, which are important regulators of ECM maturation and stiffening in the heart. Interestingly, only CsA prevented Col1a2 expression. In contrast to the other myofibroblast markers, Col3a1 was significantly increased by combined Y27 and SB treatment (Fig 4G).

For *reversing* myofibroblast gene expression, SB was the single blocker with largest inhibitory effect on myofibroblast markers (Fig 4H). In general, combinations of SB and the other blockers had a larger effect. Similar to the prevention experiment, blocker combinations with CsA were effective in reducing Col1a2 mRNA, while the combination of SB and Y27 increased Col3a1 expression (Fig 4H).

Taken together, the combination of inhibiting ROCK and TGFβRI had the strongest inhibitory effect on myofibroblast differentiation. Thus, we proceeded with this combination to study more closely the reversal of myofibroblast phenotype at different time points after isolation.

## Reversibility of myofibroblasts depends on the duration of prior culturing time on plastic

To examine whether the reversibility of myofibroblasts is lost after longer culturing time on plastic, we examined the effect of combined treatment with SB and Y27 on myofibroblasts cultured on plastic for 9 (Fig 5A) and 16 days (Fig 5B). While myofibroblasts markers Col1a1, Lox and Postn were reduced by blocker treatment at both times, Acta2, Ctgf were only affected at the early time point, suggesting that expression of these genes can only be fully reversed within the first week of culturing on plastic (Fig 5C and 5D). Furthermore, the induction of Col3a1 expression in response to inhibitor treatment only occurred at the early time point, indicating that long culture times on plastic render cardiac fibroblasts more resistant to these pharmacological inhibitors.

## Reversed cardiac fibroblasts re-differentiate on plastic but not on soft gels

To determine whether reversed cardiac fibroblasts recapitulated the mechanosensitivity and plasticity of freshly isolated resting cardiac fibroblasts, we examined the ability of reversed cardiac fibroblasts to re-differentiate on plastic after removing inhibitors, and whether this could be prevented by culturing cells on soft gels (Fig 6A). SMA fibers were present in cardiac fibroblasts on plastic and gels (Fig 6B), suggesting the presence of a myofibroblast phenotype regardless of stiffness. Despite SMA fibers and increased Acta2 mRNA, myofibroblast marker

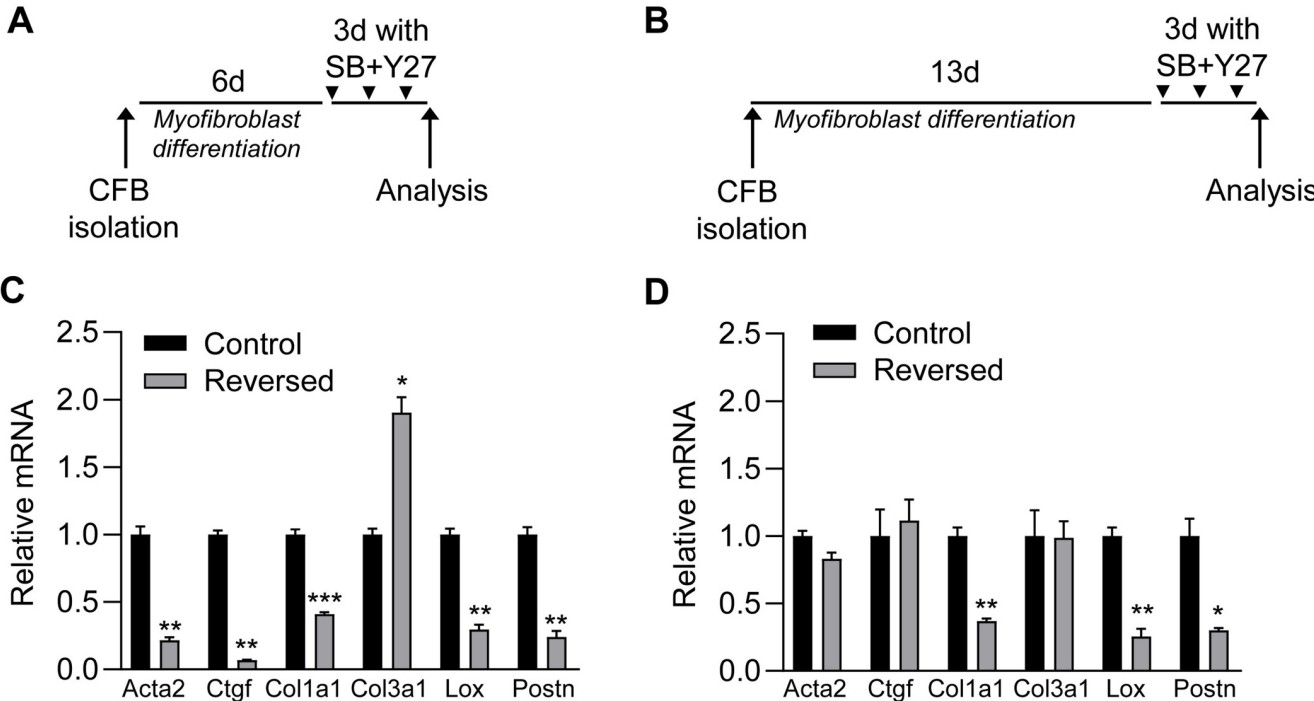

**Fig 5. Reversibility of myofibroblasts depends on the duration of prior culturing time on plastic.** A and B) Schematic illustrations of protocols used in C and D, respectively. Cardiac fibroblasts were cultured on plastic for 6 (A) or 13 (B) days (d) and then treated with blockers of ROCK (Y27) and TGFβRI (SB) for 3d. C and D) mRNA expression of myofibroblast markers actin alpha 2 smooth muscle (Acta2), connective tissue growth factor (Ctgf), collagen (Col) 1a1, Col3a1, lysyl oxidase (Lox) and periostin (Postn) in cardiac fibroblasts treated with Y27 and SB (reversed; grey bars) relative to untreated cardiac fibroblast (control; black bars). The myofibroblast marker gene measurements in D are also shown in the heat map of Fig 4H. Significant differences were determined by Student's *t*-test. N = 6 (C) and N = 3 (D).

genes Ctgf, Col1a1, Col1a2 and Col3a1 remained low when cells were cultured on soft hydrogels (Fig 6C white bars), whereas all myofibroblast markers, except Col3a1, increased during re-differentiation on plastic (Fig 6C, black bars). These results indicate that the expression of some genes, such as the collagen I genes, maintain mechanosensitivity after blocker treatment. Since Col3a1 was increased by treatment with Y27 and SB (Fig 4H), expression was not further increased during re-differentiation but rather inhibited by culturing on soft hydrogels (Fig 6C). In contrast, Acta2, Lox and Postn were increased in cardiac fibroblasts on plastic and soft gels after blocker removal, suggesting that re-activation of TGFβRI and/or ROCK signaling is occurring regardless of matrix stiffness and are strong inducers of these genes [15, 16]. Taken together, these results suggest that transient inhibition of mechanotransduction changes the mechanosensitivity of some genes.

## Discussion

*In vitro* studies of cardiac fibroblast activation are complicated by the rapid differentiation into myofibroblasts when primary cardiac fibroblasts are cultured on plastic [1]. Here we investigated whether soft hydrogels or biochemical inhibitors could maintain resting cardiac fibroblasts *in vitro*.

Although soft hydrogels initially prevented myofibroblast differentiation, SMA fibers and myofibroblast markers were eventually increased after 15 days. Myofibroblast differentiation might be induced by the high cell density at this time point. Indeed, high cell density has recently been shown to diminish the effect of hydrogel stiffness on cell phenotype. Atomic

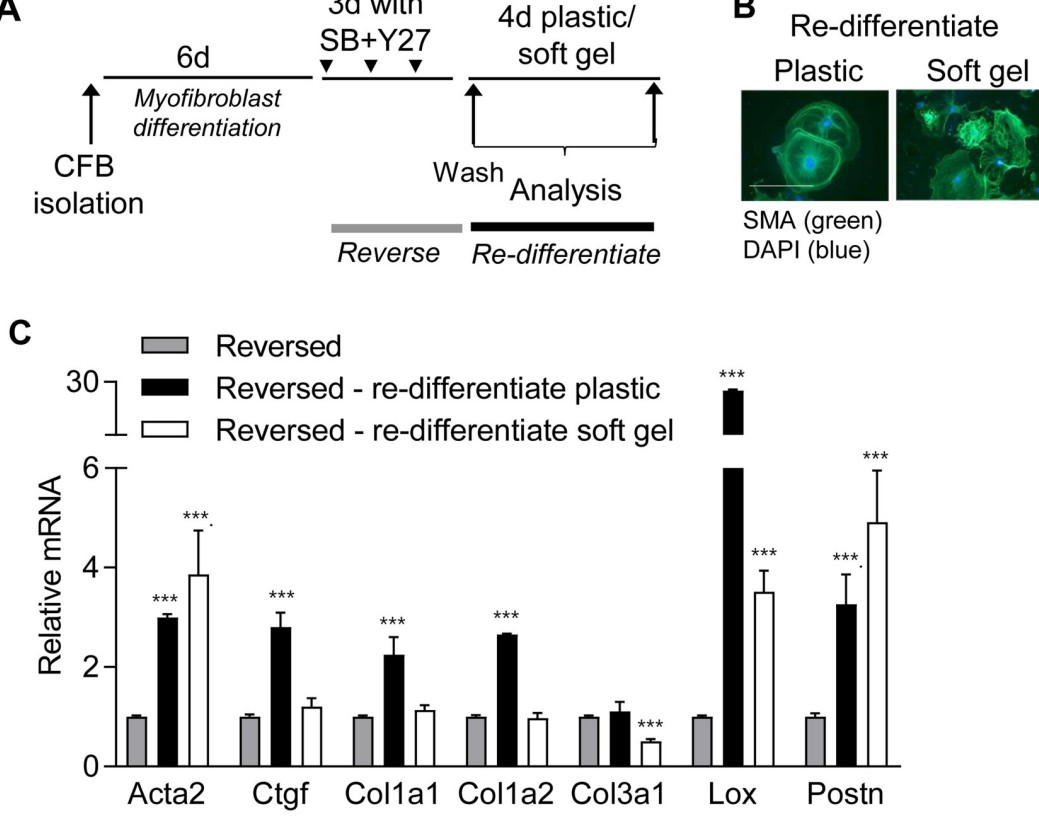

**Fig 6. Reversed cardiac fibroblasts re-differentiate on plastic.** A) Schematic illustration of re-differentiation protocol. Cardiac fibroblasts were cultured on plastic for 6 days (d) and then treated with blockers of ROCK (Y27) and TGFβRI (SB) for 3d, where after blockers were removed and cells transferred to soft gels for 4d. B) Immunostaining for smooth muscle α-actin (SMA; green). Nuclei are stained with DAPI (blue). Scale bar 100μm. C) mRNA expression of actin alpha 2 smooth muscle (Acta2), connective tissue growth factor (Ctgf), collagen (Col) 1a1, Col1a2, Col3a1, lysyl oxidase (Lox) and periostin (Postn) relative to reversed cells (grey bars), following re-differentiation on plastic (black bars) or soft gels (white bars). Significant differences were determined by Student's *t*-test with Holm-Sidak correction for multiple comparisons. N = 3.

force measurements also suggested local strain stiffening of the hydrogel when cell density is high [17]. Thus, hydrogels are most suitable for short experiments over the course of days.

It is not clear whether cardiac fibrosis and the myofibroblast phenotype can be reversed [18, 19]. TGFβ signaling is crucial for myofibroblast differentiation [20–22] and is activated by tissue stiffness due to liberation from the latent protein binding [23, 24]. ROCK is also central for the mechanosensing and response of cells [7, 25]. As anticipated, the combined use of TGFβRI and ROCK inhibitors reversed the expression of myofibroblast markers. However, SMA fibers did not disappear completely after treatment with TGFβRI and ROCK inhibitors. Others have shown that the ability of TGFβRI inhibition to reverse myofibroblast differentiation is dependent on proliferative activity [6]. Interestingly, reduced proliferation has also been associated with high SMA expression, despite reduced contraction, suggesting that Acta2 expression may also modulate cellular functions other than contractility [26].

Since prolonged culture time is necessary for many experiments (e.g. CRISPR/Cas9 gene editing), it would be of great benefit to maintain a resting cardiac fibroblast phenotype *in vitro*. Reversed cardiac fibroblasts maintained the ability to become re-activated into myofibroblasts, and this was prevented by culturing on soft hydrogels. This could provide a potential method

for studying activation of cardiac fibroblasts after initial expansion of primary cultures on plastic. One limitation to this approach is that, despite soft substrate culture conditions, SMA fibers re-appeared and Lox and Postn mRNA were increased suggesting partial myofibroblast differentiation after removing blockers.

The counter regulation of Col3a1 and Col1a1 by combined TGFβRI and ROCK inhibition was a novel finding of this study. In agreement with these results, we previously found differential effects on Col1a1 and Col3a1 expression in cardiac fibroblasts in response to substrate stiffening [5]. A shift in the ratio of collagen I/collagen III has been suggested be central for changes in myocardial stiffness [27, 28], due to the intrinsically higher stiffness of collagen I compared to collagen III fibers. Thus, Col3a1 expression might be considered beneficial in the context of fibrosis and diastolic function.

Controlling the *in vitro* environment is a challenge and because of the extensive plasticity of cardiac fibroblasts, variation in culturing conditions, apart from tissue stiffness, might affect cell phenotype. These include effects of serum composition, substrate protein coating, and number of cell passage. Based on our results, we recommend using blocker-induced reversal of myofibroblasts in combination with soft hydrogels to study the mechanical induction of collagen I and III expression, while other features of myofibroblasts such as SMA fibers, Lox and Postn should be studied in freshly isolated cardiac fibroblasts.

## Acknowledgments

We are highly grateful to Volkan Turan and Jennifer Stowe for their excellent technical assistance.

## Author Contributions

**Conceptualization:** Andrew D. McCulloch, Cord H. Brakebusch, Kate M. Herum.

**Data curation:** George Gilles, Kate M. Herum.

**Formal analysis:** George Gilles, Kate M. Herum.

**Funding acquisition:** Andrew D. McCulloch, Kate M. Herum.

**Investigation:** George Gilles, Kate M. Herum.

**Methodology:** George Gilles, Kate M. Herum.

**Project administration:** George Gilles, Kate M. Herum.

**Resources:** Cord H. Brakebusch.

**Supervision:** Andrew D. McCulloch, Cord H. Brakebusch, Kate M. Herum.

**Validation:** Kate M. Herum.

**Visualization:** Kate M. Herum.

**Writing – original draft:** George Gilles, Kate M. Herum.

**Writing – review & editing:** George Gilles, Andrew D. McCulloch, Cord H. Brakebusch, Kate M. Herum.

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
