## [Decision Letter · Decision Letter 0]

24 Jun 2020

PONE-D-20-17115

Maintaining resting cardiac fibroblasts *in vitro* by disrupting mechanotransduction

PLOS ONE

Dear Dr. Herum,

Thank you for submitting your manuscript to PLOS ONE. After careful consideration, we feel that it has merit but does not fully meet PLOS ONE’s publication criteria as it currently stands. Therefore, we invite you to submit a revised version of the manuscript that addresses the points raised during the review process.

The reviewer identified several technical and conceptual areas that require revision. In particular, the study is largely descriptive without new insights into mechanistic relationships between culture conditions and outcomes. Most of the documented relationships as a function of culture conditions have been described elsewhere. Identification of molecular interactions that drive the differences in cell behavior or gene regulation as a function of culture conditions would provide new information to advance the field. The reviewer has provided a thorough identification of additional changes that will be required before the manuscript can be accepted.

We look forward to receiving your revised manuscript.

Kind regards,

Philip C. Trackman, Ph.D.

Academic Editor

PLOS ONE

Journal Requirements:

2. Please upload a copy of Figure 3, to which you refer in your text on page 15, 16 and 17 (Figure 3 cited, but Supplementary Figure 3 uploaded).  If the figure is no longer to be included as part of the submission please remove all reference to it within the text.

[A.D.M. is a co-founder of and has an equity interest in Insilicomed Inc. and an equity interest in Vektor Medical, Inc.. He serves on the scientific advisory board of Insilicomed, and as scientific advisor to both companies. Some of his research grants have been identified for conflict of interest management based on the overall scope of the project and its potential benefit to these companies. The author is required to disclose this relationship in publications acknowledging the grant support; however, the research subject and findings reported in this study did not involve the companies in any way and have no relationship with the business activities or scientific interests of either company. The terms of this arrangement have been reviewed and approved by the University of California San Diego in accordance with its conflict of interest policies.]

Reviewers' comments:

Reviewer's Responses to Questions

**Comments to the Author**

1. Is the manuscript technically sound, and do the data support the conclusions?

Reviewer #1: Partly

2. Has the statistical analysis been performed appropriately and rigorously? 

Reviewer #1: I Don't Know

3. Have the authors made all data underlying the findings in their manuscript fully available?

Reviewer #1: No

4. Is the manuscript presented in an intelligible fashion and written in standard English?

Reviewer #1: No

5. Review Comments to the Author

Reviewer #1: To understand the effect of mechanotransduction on fibroblasts to myofibroblasts differentiation of cardiac fibroblasts, it is highly critical to maintain the resting fibroblasts that are freshly isolated from tissues. The authors screened in vitro culturing conditions to slow down activation/differentiation of cardiac fibroblasts to myofibroblasts and recommend adding inhibitors specific to TGFβR1 and ROCK as well as seeding fibroblasts on soft hydrogels.

It is an important study, however, the manuscript needs to be revised thoroughly to address the following concerns.

Major issues

1. It is well established that TGFβR1 and ROCK play critical roles in fibroblasts to myofibroblasts differentiation and inhibitors of these signaling pathways inhibit the differentiation. The results described in this manuscript do not add anything substantially new to our current understandings. It seems that once fibroblasts are activated through mechanotransduction by ECM stiffening to form SMA fibers, SMA fibers persist even after transferring cells onto a soft hydrogel or treatment with inhibitors of TGFβR1 and ROCK, therefore changes in cell morphology are negligible. LOX and LOX-family of proteins are upregualted via TGFβR1 signaling pathways, therefore upregulation of LOX gene was detected when TGFβR1 blocker was removed in Panel A and C (Figure 5). Authors need to cite literature concerning LOX gene and TGFβ.

2. It is better to include a schematic diagram for relevant signaling pathways for fibroblasts to myofibroblasts differentiation.

3. The authors need to provide rational/explanation for why they examine the effect of calcineurin inhibitor cyclosporine A in this study.

4. Images (Panel C and D) in Figure 3 (labeled as Supllemantary Figure 3 in the manuscript) should be replaced with images in better resolution.

5. It is intriguing that LOX is highly upregulated when cells are cultured on a plastic (Figure 1, Panel F and Figure5, Panel C).The authors need to add their explanation/interpretation for this.

6. Figure legends in this manuscript need to be revised thoroughly. At the current state, it is not self explanatory.

Figure 1.

- Which data of Panel C do have N= 3 and which data of Panel C have N= 8?

- Which data (day) do correspond to Panel F? What is Y-axis in Panel F?

- Panel A and C, it is better to include more data points between day 9 and day 15 so that the critical time point for differentiation can be defined.

- Panel D describes fibroblasts cultured 9 days on plastic and switched to soft hydrogel and cultured for additional 4 days, but this is not explained in the legend. Is it necessary to incubate 9 days to see full differentiation? Some explanation for this experimental set-up needs to be provided.

- The text of this manuscript mentions Figures 2E and 2F but are actually Figures 1E and 1F, respectively.

Figure 2.

- The label “anti-SMA” in Panel A, B, C should be SMA.

- What are the differences between Panel B 50 μM and Panel C 50 μM?

Supplementary Figure 3.

- This is called Supplementary Figure 2 within the text, but it should be Figure 3.

- Panel A. “wth” should be “with”

- Panel C and D. Description of upper and lower panels for effect of blockers need to be provided.

Figure 4.

- Panel A and B, provide detailed procedures, e.g. Fibroblasts were cultured on plastic for 6 days (Panel A) and 13 days (Panel B).

- Panel C and D should include relative mRNA levels of these markers before addition of SB+Y27.

Other issues

1. There are too many non-essential and introductory elements in a sentence and that makes this manuscript difficult to follow.

2. Abstract

The last sentence, starting with Importantly, needs to be revised to address the conclusion of this paper and the message should be clearer.

3. Introduction

Line 6 from the top, lysyl oxidase-like 2 (LOXL2) should be added as LOXL2 has also shown to play critical role in cardiac interstitial fibrosis by Yang, J. et al Nature Comm. 2016, 7, 13710.

4. Results

The following sentences need to be revised for better understanding.

under Relieving cells of high stiffness delays, but does not prevent myofibroblast differentiation.

Line 2 in the second paragraph, “SMA staining intensity was low for cardiac fibroblasts cultured on soft gels for 3 and 9 days, but increased after 15 days on gels (Figure 1B), at which time cells were highly confluent.” � SMA staining and confluency between day 9 and day 15 should be included (e.g. day 11, day 13).

Line 6 in the second paragraph,

“Marker expression was no longer suppressed by soft compared with stiff culture substrates.”

“for the reverse experiment, SB inhibition, as well as all combinations of blockers, reduced SMA staining intensity, albeit the effect of blocker combinations were more pronounced.”

Minor

1. Full name and abbreviations.

Abstract:

Line 8 from the top, (SMA) should be inserted after smooth muscle α-actin.

Line 10 from the top, transforming growth factor β receptor I should be inserted before (TGFβRI) as well as Rho-associated protein kinase before (ROCK). Actin alpha2 smooth muscle gene should be inserted before (Acta2).

Introduction:

Line 4 from the top, full name for Acta 2 should be given.

Line 9 from the top, full name for Tcf21 should be given.

2. Some errors listed below need to be revised.

Abstract:

Line 4 from the top, “It is therefore challenging to study resting cardiac fibroblasts and their activation, in vitro.” The comma should be removed.

Line 6 from the top, “maintain resting cardiac fibroblasts for up to 5 days but some experiments” Insert comma before but

Introduction:

line 5 from the bottom, “However, the use of soft hydrogels have limitations” � “The use of soft hydrogels has some limitations”

Results:

Under “Inhibiting ROCK and TGFβRI is most efficient for preventing and reversing the myofibroblast phenotype” on the second page,

In the paragraph starting with “For the reverse experiment, “

“blocker combinations where most efficient in preventing and reversing” should read as “blocker combinations were most efficient at preventing and reversing”

in the paragraph starting with “We next measured…”

“only SB prevented Acta2 upregulation confirming that” Insert a comma after upregulation.

6. PLOS authors have the option to publish the peer review history of their article (what does this mean?). If published, this will include your full peer review and any attached files.

Reviewer #1: No

---

## [Author Response · Author response to Decision Letter 0]

3 Sep 2020

We have responded to all specific reviewer and editor comments in the "Response to Reviewer".

---

## [Decision Letter · Decision Letter 1]

23 Sep 2020

PONE-D-20-17115R1

Maintaining resting cardiac fibroblasts *in vitro* by disrupting mechanotransduction

PLOS ONE

Dear Dr. Herum,

Thank you for submitting your manuscript to PLOS ONE. After careful consideration, we feel that it has merit but does not fully meet PLOS ONE’s publication criteria as it currently stands. Therefore, we invite you to submit a revised version of the manuscript that addresses the points raised during the review process.

The expert reviewer clearly appreciated the improvement in the manuscript. A few adjustments in the writing and a couple of clarifications are still needed, which would further improve the manuscript. Once these have all been addressed, then the manuscript will likely be suitable for publication. I understand that the reviewer probably did not find the high resolution figures available by clicking the link located at the upper right corner of each figure on the PLoS website, so your figures are fine as is with respect to resolution. 

We look forward to receiving your revised manuscript.

Kind regards,

Philip C. Trackman, Ph.D.

Academic Editor

PLOS ONE

Reviewers' comments:

Reviewer's Responses to Questions

**Comments to the Author**

1. If the authors have adequately addressed your comments raised in a previous round of review and you feel that this manuscript is now acceptable for publication, you may indicate that here to bypass the “Comments to the Author” section, enter your conflict of interest statement in the “Confidential to Editor” section, and submit your "Accept" recommendation.

Reviewer #1: All comments have been addressed

2. Is the manuscript technically sound, and do the data support the conclusions?

Reviewer #1: Yes

3. Has the statistical analysis been performed appropriately and rigorously? 

Reviewer #1: Yes

4. Have the authors made all data underlying the findings in their manuscript fully available?

Reviewer #1: Yes

5. Is the manuscript presented in an intelligible fashion and written in standard English?

Reviewer #1: Yes

6. Review Comments to the Author

Reviewer #1: The authors made efforts to address my concerns and the manuscript reads significantly better.

However, there are some issues that need to be sorted out before considering this manuscript for publication on this journal.

Major problem:

I am not sure why images are still at low resolution in this revision, although authors stated that they replaced older images with images in higher resolution. Perhaps it is due to how this PDF version of article was generated. Figures and legends are fuzzy with some background. Immunostaining images are not in publishable quality.

1. The authors response to the first critique under Major issues, “It is well established that TGFβR1 and ROCK play critical roles in fibroblasts to myofibroblasts differentiation and inhibitors of these signaling pathways inhibit the differentiation. The results described in this manuscript do not add anything substantially new to our current understandings.” is reasonable.

“Although it is known that TGFβRI and ROCK are important for myofibroblast differentiation,

their relative contributions and the effect of dual inhibition have not previously been tested.”

- This provides the rational for the research conducted in this study and should be included in the introduction.

“Our results bring novel information regarding the relative importance of these specific signaling

pathways in regulating different myofibroblast-associated genes. E.g. the finding that col3a1 was upregulated by dual inhibition was a novel finding of this paper. Furthermore, combining blocker treatment with the use of soft hydrogels has not been tested previously and revealed that only some genes remain mechano-sensitive after phenotypic reversion. This is important information for researchers using inhibitors to maintain a “resting” cardiac fibroblast phenotype in vitro as well as for our current understanding of mechanosensitive gene regulation.”

- This part should be included in abstract and discussion/conclusion.

2. Abstract Line 11-12 “Myofibroblast differentiation was prevented after 3 and 9 days on soft hydrogels. However, after 15 days, myofibroblasts appeared.”

“after 3 and 9 days on soft hydrogels” �”at 3 and 9 days after transferring to soft hydrogels”

3. Figure 2

Figure legend should read, “mRNA expression of myofibroblast markers and Tcf21 at 13 days after seeding. Y-axis value, 2-∆Ct, represents mRNA levels for each gene relative to 18S rRNA. ∆Ct is the Ct value of the gene of interest minus the Ct value of 18S rRNA.”

The following explanation should be included in text. “One Ct value reflects a doubling of mRNA and Ct values are transformed to a linear scale by exponentiation. Since Ct levels are inversely proportional to the amount of target nucleic acid in the sample, ∆Ct is a negative value. The data presented in Panel C where first transformed using 2-∆Ct and then presented as relative to the control group to clearly show the effect of soft gels for each gene.”

The difference between Figure 2C (little effect between soft gel and plastic) and Figure 2F (very high difference after transferring to soft gel) is striking. Could this be related to confluency of cells? It seems unlikely that cells transferred to soft gels kept the exact same confluency as those kept on plastic.

4. Figure 4

Panels A and B should have days on the arrows similar to Figure 2.

5. Figure 5

Figure legend for panel B is missing. The authors need to include some explanation/discussion about the difference in the level of Col3a1. Why it is 2X increased for reversed in panel C but no effect in D?

7. PLOS authors have the option to publish the peer review history of their article (what does this mean?). If published, this will include your full peer review and any attached files.

Reviewer #1: No

---

## [Author Response · Author response to Decision Letter 1]

30 Sep 2020

We have addressed all specific comments in the attached rebuttal letter and in the revised manuscript.

---

## [Decision Letter · Decision Letter 2]

14 Oct 2020

Maintaining resting cardiac fibroblasts *in vitro* by disrupting mechanotransduction

PONE-D-20-17115R2

Dear Dr.Herum,

We’re pleased to inform you that your manuscript has been judged scientifically suitable for publication and will be formally accepted for publication once it meets all outstanding technical requirements.

Kind regards,

Philip C. Trackman, Ph.D.

Academic Editor

PLOS ONE

Additional Editor Comments (optional):

Reviewers' comments:

Reviewer's Responses to Questions

**Comments to the Author**

1. If the authors have adequately addressed your comments raised in a previous round of review and you feel that this manuscript is now acceptable for publication, you may indicate that here to bypass the “Comments to the Author” section, enter your conflict of interest statement in the “Confidential to Editor” section, and submit your "Accept" recommendation.

Reviewer #1: All comments have been addressed

2. Is the manuscript technically sound, and do the data support the conclusions?

Reviewer #1: Yes

3. Has the statistical analysis been performed appropriately and rigorously? 

Reviewer #1: Yes

4. Have the authors made all data underlying the findings in their manuscript fully available?

Reviewer #1: Yes

5. Is the manuscript presented in an intelligible fashion and written in standard English?

Reviewer #1: Yes

6. Review Comments to the Author

Reviewer #1: The authors adequately addressed all my concerns. This manuscript is ready for publication on PLOS ONE.

7. PLOS authors have the option to publish the peer review history of their article (what does this mean?). If published, this will include your full peer review and any attached files.

Reviewer #1: No

---

## [Editor Report · Acceptance letter]

16 Oct 2020

PONE-D-20-17115R2 

Maintaining resting cardiac fibroblasts *in vitro* by disrupting mechanotransduction 

Dear Dr. Herum:

I'm pleased to inform you that your manuscript has been deemed suitable for publication in PLOS ONE. Congratulations! Your manuscript is now with our production department. 

Kind regards, 

on behalf of

Dr. Philip C. Trackman 

Academic Editor

PLOS ONE